# A Kirkpatrick Model Process Evaluation of Reactions and Learning from My Strengths Training for Life™

**DOI:** 10.3390/ijerph191811320

**Published:** 2022-09-08

**Authors:** Mary L. Quinton, Grace Tidmarsh, Benjamin J. Parry, Jennifer Cumming

**Affiliations:** 1School of Sport, Exercise and Rehabilitation Sciences, University of Birmingham, Birmingham B15 2TT, UK; 2Institute for Mental Health, University of Birmingham, Birmingham B15 2TT, UK

**Keywords:** young people experiencing homelessness, disadvantaged youth, engagement, community-based research, positive youth development, mental skills training

## Abstract

Underpinned by the New World Kirkpatrick model, and in the context of a community-based sport psychology programme (My Strengths Training for Life™) for young people experiencing homelessness, this process evaluation investigated (1) young peoples’ reactions (i.e., program and facilitator evaluation, enjoyment, attendance, and engagement) and learning (i.e., mental skills and transfer intention), (2) the relationship between reaction and learning variables, and (3) the mediators underpinning this relationship. A total of 301 young people living in a West Midlands housing service completed questionnaires on demographics and reaction and learning variables. Higher levels of programme engagement were positively associated with more favourable reactions to the programme. Enjoyment positively predicted learning outcomes, which was mediated by transfer intention. Recommendations are made for (1) a balance between rigor and flexibility for evaluation methods with disadvantaged youth, (2) including engagement as well as attendance as indicators of meaningful programme participation, (3) measuring programme experiences (e.g., enjoyment) to understand programme effectiveness, and (4) providing opportunities for skill transfer during and after programme participation. Our findings have implications for researchers, programme commissioners, and policymakers designing and evaluating programmes in community-based settings.

## 1. Introduction

Young people experiencing homelessness are a unique population within the community, with bespoke needs to face numerous and complex economic, health, and social challenges and inequalities [1,2]. Research has often been conducted *on* this subpopulation, but less frequently *with* these young people; as a result, this marginalised group is underrepresented in research [3,4]. In a systematic review of strategies to increase health research within socially disadvantaged groups, Bonevski et al. [5] noted that research should operate via community partnerships to increase these groups’ representation. However, only 4 of 116 studies in this review included people experiencing homelessness. For research with disadvantaged young people to be representative, the onus needs to shift from labelling this group as “hard to reach” to increasing researchers’ responsibility to create accessible opportunities for engaging with these groups [4]. Therefore, researchers should work closely with community collaborators to gain young people’s input into programme design and evaluation [1], which would enhance the relevance of subsequent programmes to young people’s needs, and result in more tailored and effective policy development [6].

### 1.1. The My Strengths Training for Life™ Program

Historically, there have been few evidence-based interventions with young people experiencing homelessness [7]. As part of a larger community-based participatory research project [8], this article describes a quantitative process evaluation of My Strengths Training for Life (MST4Life)™, delivered in partnership with a housing service for young people experiencing homelessness. This research was underpinned by strengths-based psychology and positive youth development (PYD), where core components include a focus on assets over deficits, providing meaningful opportunities to develop and build upon existing mental strengths, and promoting positive and healthy adult and peer relationships [8]. PYD and mental skills training programmes (e.g., MST4Life™) are centred around the development of mental skills such as intentional self-regulation, problem solving, and emotional regulation [3,9]. Research has demonstrated the effectiveness of MST4Life™ for promoting psychosocial development, intentional self-regulation, and integrating young people back into society (see Figure 1 for an updated logic model based on evidence to date). However, a research gap with this population is a quantitative assessment of how young people’s experiences of strengths-based programmes can lead to learning outcomes, which would contribute to the small evidence base within this area.

### 1.2. Process Evaluations

One way to ascertain young people’s views on their experiences of participating in programmes is through process evaluations. Process evaluations seek to understand an intervention’s context (e.g., environment), mechanisms (e.g., participants’ responses to, and interactions with, the intervention, mediators, and unintended consequences), and implementation (e.g., delivery quality and quantity) [17]. This type of evaluation ensures that young people’s views are appropriately captured, and that service provision is meeting actual rather than perceived needs [6,17,18]. It is well evidenced that a flexible, tailored approach is needed for research with young people experiencing homelessness, and the strict criteria and positivist approach of randomised controlled trials are often not possible or appropriate (e.g., due to the transient nature of the population) [1,9,18,19]. Therefore, it is important that programme evaluators select research methods and evaluation models that provide the opportunity for young people’s views of that programme to be represented in research [14,19].

Limited process evaluations have been conducted on programmes with young people experiencing homelessness [1,6]. Addressing one aspect of process evaluation, Krabbenborg et al. [7] assessed the fidelity of Houvast—a Danish strengths-based intervention with young people experiencing homelessness—and reported mixed findings in Houvast’s implementation across shelters. In MST4Life™, different components of process evaluations have been conducted, including a feasibility study [11], realist evaluation [12], and fidelity of strengths-based delivery style with frontline staff [15]. In another study, Tidmarsh et al. [14] conducted a qualitative process evaluation of the implementation of MST4Life™, using diary rooms to explore young people’s perceptions of barriers to and enablers of engagement. A key theme was the importance of using engaging activities to facilitate the development of mental skills that could be used within and outside of MST4Life™ (i.e., skill transfer). Despite these MST4Life™ process evaluations, it is still not clear what mechanisms are underpinning young people’s reactions to programmes (e.g., enjoyment), or how these reactions, in turn, lead to learning outcomes (e.g., mental skills developed and transfer intention to other settings), whereby doing so would provide support for the programmes’ logic model [8] (Figure 1). Although the diary room is flexible and provides a platform for participants’ voices [14], a quantitative process evaluation would offer complementary information by overcoming the limitations of the diary room (e.g., not all young people may be comfortable with sharing views in this way) and ensure that these young people’s views are still captured within such a heterogeneous population [9]. An overall mixed-methods process evaluation approach to MST4Life™ also meets recommendations for improving the scope and understanding of process evaluations in disadvantaged youth [6,17].

### 1.3. The New World Kirkpatrick Model

A well-known model of evaluation that provides structure to investigate participants’ engagement and learning is the New World Kirkpatrick model [20]. The model has five levels: (1) reaction (i.e., participants’ responses to the programme), (2) learning (i.e., the extent to which the participants obtained the learning outcomes), (3) behaviour (i.e., behavioural changes from participating), (4) results (i.e., the impact of the programme on wider organisational goals), and (5) return on expectations (i.e., the extent to which the collaborator expectations were met) [21]. The original model was outcome-focused [22], but the New World model is suitable for process evaluations, as it emphasises the importance of processes and the impact of learner characteristics on programme outcomes [20,23,24]. Although this model originated from business, it has since been applied to diverse settings such as outdoor adventure education (OAE) [25] and nursing [26]. However, it has not yet been used to underpin evaluation in the context of young people experiencing homelessness, despite the potential value of the model in providing a framework to allow consistency across interventions and clearer recommendations on programme design, delivery, and evaluation for researchers, policymakers, and programme commissioners.

The New World Kirkpatrick model proposes that level 1 (reaction) should measure participants’ engagement in the learning experience [20]. Research has predominantly measured attendance as an indicator of programme success, and has overlooked engagement [19]. Engagement focuses on the quality of experience while involved in programme activities, and plays a key role in recruiting and retaining participants—especially for older youth, who have more choices in how they spend their time [27]. In health, educational, and outreach settings, better engagement in programmes is associated with better outcomes, such as higher grades or greater wellbeing and functioning [28,29,30]. Young people demonstrate their engagement through observable behaviours (e.g., contributing to discussions) and displaying a positive attitude towards activities [31]. While those experiencing homelessness can be reluctant to engage in programmes—for instance, due to inaccessibility, or lack of trust in service providers [32]—programmes that prioritise engagement build relevance for participants, and may help to overcome perceived obstacles to participation [31]. PYD programs such as MST4Life™ that focus on relationships with others, skill-building opportunities, and prioritise strengths over deficits, are likely to promote engagement (e.g., active contributions, engaging with others), and this should therefore be reflected in their evaluation. From a process evaluation perspective, measuring engagement would also provide an indicator of intrinsic motivation (i.e., young people attend for their own benefit rather than being motivated by external reasons) [11]. Therefore, the present study included engagement alongside attendance as measures of reaction, and examined their relationship with learning variables (i.e., mental skills used and transfer intention). If attendance and engagement are associated with learning outcomes, this would reinforce both variables as important indicators to measure when delivering and evaluating programmes with this population.

Level 2 (learning) of the New World Kirkpatrick model is proposed to evaluate the extent to which learning has been achieved. Extending this idea further, Cooley et al. [23] found that learning and its transfer to other settings are influenced not only by reaction variables, but also by contextual and learner characteristics. Transfer of learning should be an important goal for psychoeducational programmes. It is often assumed that learning (i.e., the knowledge and skills developed and used in a controlled context, such as school or MST4Life™) will automatically transfer outside of this context to other areas of life, but research shows that transfer of learning can be limited [33]. It is therefore important to intentionally create relevant opportunities for transfer within such contexts so that transfer opportunities in real-world settings can be more easily identified when presented [33]. Although there has been mixed evidence to support causal links between Kirkpatrick evaluation levels [24], Cooley et al. [25] showed that reaction variables (e.g., program enjoyment and satisfaction) predicted learning (e.g., improved group-work skills and intention to transfer) in university students participating in an OAE course. Importantly, intention to transfer learning has been noted as a vital prerequisite to learning transfer [23]. Referring to the logic model (Figure 1), one of the expected outcomes of MST4Life™ is transfer of learning [8]. Therefore, building from the work of Cooley et al. [23,25] and Kirkpatrick and Kirkpatrick [20], the current process evaluation investigated whether reactions (e.g., programme evaluation and enjoyment, facilitator evaluation) predicted learning (e.g., mental skills experiences in MST4Life™), but also whether this relationship was mediated by intention to transfer these skills. If supported, this relationship would hold significant implications for programmes with young people experiencing homelessness, indicating that opportunities for skill transfer should continue once the programme has ended, so as to encourage transfer to other contexts (e.g., education, employment, or training (EET)) [23].

### 1.4. Study Aims and Hypotheses

Underpinned by the New World Kirkpatrick model [20], and building upon qualitative MST4Life™ research [12,14], the first aim of this process evaluation was to investigate young people’s reactions (i.e., programme evaluation, enjoyment, facilitator evaluation, attendance, and engagement) to and learning from MST4Life™ (i.e., mental skills used and transfer intention). The second aim was to explore the relationship between reaction and learning variables, with the third aim being to determine the mediators underpinning this relationship. Based on the work of Cooley et al. [23], transfer intention was included as a mediator as well as an indicator of learning, as it was also noted as a vital prerequisite to learning transfer. It was hypothesised that (1) young people would have favourable reactions to and learning from MST4Life™ due to taking part in a PYD programme [3], (2) reaction variables would be positively correlated with learning variables [25], and (3) transfer intention would mediate the relationship between reaction variables and mental skills developed in MST4Life™ [23]. The higher levels (i.e., behaviour, results, and return on expectations) of the Kirkpatrick model are not reported in this study, but have been reported elsewhere [12,16].

This study makes an original contribution by (1) extending the scant literature on process evaluations with young people experiencing homelessness, (2) measuring engagement in the programme in addition to attendance, and (3) applying the New World Kirkpatrick model in a youth homelessness context. The findings could have important implications for housing services, researchers, programme commissioners, and policymakers working with this population, providing recommendations for conducting process evaluations in community-based settings, as well as what measures are important to consider (e.g., attendance vs. engagement; transfer intention).

## 2. Materials and Methods

### 2.1. Participants

The sample consisted of 301 young people (*M* age = 19.64, *SD* = 2.31) supported by the housing service. A breakdown of the participants’ demographic information can be found in Table 1. Through a purposive sampling approach led by the housing service staff, the inclusion criteria were that the young people (a) lived in supported accommodation or a floating support service, (b) engaged in at least one MST4Life™ session, and (c) were either recruited on the basis of being not currently engaged in meaningful activity (e.g., EET, volunteering, apprenticeships), or were considered by staff to be likely to benefit from the programme with regards to their potential for developing mental skills [8]. As the programme was strengths-based and very inclusive in nature, there were no set exclusion criteria; however, there were various reasons why young people did not or were not able to attend (e.g., session time clashed with work/college).

### 2.2. Intervention

MST4Life™ is a community-based PYD programme that helps young people to recognise existing strengths and self-regulate their thoughts, feelings, and behaviours, with the intention to transfer skills into other settings (e.g., EET) [9]. MST4Life™ was delivered within a large West Midlands housing service, with a psychologically informed environment (PIE) organisational approach [8]. Young people also had opportunities to engage in other activities for self-development, such as life skills workshops (e.g., cooking and budgeting) and a youth advocacy group.

The core principles of MST4Life™, as determined by young people and staff from the housing service, were fun and interactive, flexible, and young-person-led [8]. The programme consisted of 10 sessions that took place at the supported accommodation sites (average duration 2 h) or in the community (4 h). For more information about programme activities, see Cooley et al. [10] and Cumming et al. [8]. The programme also included an OAE residential, but these data are reported elsewhere [12].

The programme facilitators’ backgrounds were primarily in sport psychology. Aligned with the sport psychology underpinning of MST4Life™, self-determination theory (SDT) grounded the facilitators’ approach [34], supporting basic psychological needs for autonomy, competence, and relatedness [11]. The latter particularly encouraged rapport development (e.g., facilitators welcomed each young person, and engaged in informal conversation to get to know people better), which was considered to be vital for working with young people with complex psychological needs. To further adapt to working with these young people, the facilitators completed training courses on PIE, motivational interviewing, and mental health first aid. The facilitators also engaged in reflective practice with the housing services’ clinical psychologist.

### 2.3. Measures

#### 2.3.1. Demographics

Young people self-reported their gender, ethnicity, social inclusion status, and learning difficulties (Table 1). Guidelines from appropriate resources were followed to ensure that the categories were suitably named [9,35,36]. 

#### 2.3.2. Reactions

##### Attendance

Attendance was recorded for each session. The maximum possible attendance was 10 sessions. An average attendance score was created for the overall sample. 

##### Engagement

Engagement was rated by facilitators for each session on a scale of 1 (*not at all engaged*) to 10 (*could not be more engaged*). As most sessions were delivered by two facilitators, an average score was taken across their two scores. Then, an average score was created for each young person across the number of sessions attended. 

##### Programme Evaluation

Programme evaluation was measured using a two-item index [37], with wording adapted for the current research (e.g., “Overall, the MST4Life™ program was excellent”). Participants rated the extent to which they agreed with each statement on a scale of 1 (*not at all true*) to 5 (*very true*), with an average score created across the two items. 

##### Facilitator Evaluation 

Using a two-item index [37], participants rated the extent to which they agreed with the statements (e.g., “Overall, the MST leader was excellent”) on a Likert-type scale from 1 (*not at all true*) to 5 (*very true*). An average score was created across the two items.

##### Programme Enjoyment

Enjoyment of MST4Life™ was assessed through four items adapted from the Intrinsic Motivation Inventory [38], such as “The activities were fun to do”. Participants rated the extent to which they agreed with each statement on a Likert-type scale from 1 (*not at all true*) to 5 (*very true*), with an average score created across the four items. Cooley et al. [25] used these three measures as indicators of reaction, and found them to be reliable [25]. 

#### 2.3.3. Learning 

##### Transfer Intention 

This questionnaire consisted of four items adapted from Cooley et al. [25] to assess participants’ intentions to transfer mental skills after MST4Life™. Participants rated their intentions on a Likert-type scale from 1 (*extremely unlikely*) to 5 (*extremely likely*), with an example item being “I plan to use the mental skills I developed in the future”. An average score was created across the four items, with items previously shown to be reliable [25].

##### Mental Skills Experiences

Measured using the Youth Experience Survey 2.0 (YES-2) [39], participants rated their perceived opportunities to develop mental skills on a Likert scale from 1 (*not at all*) to 4 (*yes, definitely*). For this study, only the following subscales were assessed, with a total of 21 items: goal-setting (e.g., “I set goals for myself in this activity”), effort (e.g., “I put all of my energy into this activity”), problem solving (e.g., “I learned about developing plans for solving problems”), time management (e.g., “I learned about organising time and not procrastinating), emotional regulation (e.g., “I learned that my emotions affect how I perform”), and group work (e.g., “I learned to be patient with other group members”). An average score was created for each subscale. Previous research has found these subscales to be reliable when administered to MST4Life™ participants [9]. 

### 2.4. Procedure

Ethical approval was obtained from the university’s ethics committee (ERN_21-1017). Participants were informed about the research verbally and through an information sheet to ensure their understanding. Informed consent was obtained prior to completion of the questionnaires. All participants used ID numbers or pseudonyms instead of their real names when completing the questionnaires to maintain their anonymity. Facilitators explained the questionnaire and encouraged the participants to complete it as honestly as possible, emphasising that there were no right or wrong answers. Facilitators also provided help to those who found it difficult to understand or read the questionnaires. Data were collected between October 2014 and June 2019 and at two time points in the programme: Session 2 (demographics) and Session 10 (reactions and learning).

### 2.5. Data Screening and Analyses

Data were screened and cleaned in accordance with the recommendations of Tabachnick and Fidell [40]. Cronbach alphas can be found in Table 2. Univariate and multivariate outliers were determined by inspecting z-scores (< or >3.29) and the Mahalanobis distance at *p* < 0.001, respectively. The randomness of missing data was determined by Little’s missing completely at random (MCAR) test [41]. The Benjamini–Hochberg correction was implemented to control for multiple comparisons and reduce type 1 error by adjusting the false discovery rate [42]. This method has been used previously in sport psychology research to maintain statistical power when using alpha adjustments [43,44].

Preliminary analyses consisted of one-way ANOVAs and MANOVAs to investigate demographic differences (i.e., gender, ethnicity, social inclusion, learning difficulties) in the dependent variables (i.e., reaction and learning). The main analyses consisted of Pearson’s bivariate correlations to investigate the relationships between study variables. A series of hierarchical linear regressions were conducted to determine the extent to which reaction variables (i.e., attendance, engagement, programme evaluation, facilitator evaluation, programme enjoyment) predicted learning (i.e., mental skills and transfer intention). Due to the heterogeneity of the sample [2,9], demographics were entered in the first step to control for variance. Collinearity diagnostics were checked to ensure that there was no evidence of multicollinearity (VIF < 10; tolerance > 0.10) [45]. Mediation analyses were carried out through testing for indirect effects via the PROCESS add-on in SPSS (model 4) [46]. Variables that were not significant predictors in the linear regressions were not considered in mediation analyses [47]; therefore, programme enjoyment was the predictor, transfer intention was the mediator, and mental skills were the outcome variables. Separate tests were run for each outcome variable at a 90% confidence interval, generated from bootstrapping of 5000 samples. Effect sizes were reported using the completely standardised indirect effect [48].

## 3. Results

### 3.1. Preliminary Analyses

#### 3.1.1. Data Screening and Cleaning 

Univariate outliers were identified for three reaction items: “I would recommend the MST program to a friend” (*z* = −4.15), “Overall, the MST leader was excellent” (*z* = −4.03), and “I would recommend this MST leader to a friend” (*z* = −4.32). These outliers were retained to reflect the full range of programme feedback within the data. There were three multivariate outliers identified through inspection of the Mahalanobis distance and subsequently removed. Any missing data were MCAR according to Little’s test (*p* > 0.05).

#### 3.1.2. Sample Descriptors

The average scores were 5.41 (*SD* = 2.68) for attendance, 8.24 (*SD* = 1.11) for engagement, 4.40 (*SD* = 0.75) for programme evaluation, 4.57 (*SD* = 0.66) for facilitator evaluation, 4.32 (*SD* = 0.82) for programme enjoyment, and 3.96 (*SD* = 0.89) for transfer intention. The mental skill that was most developed over MST4Life™ was effort (*M* = 3.18; *SD* = 0.65), followed by group work (*M* = 3.13; *SD* = 0.55), problem solving (*M* = 3.10; *SD* = 0.67), time management (*M* = 2.99; *SD* = 0.69), goal-setting (*M* = 2.98; *SD* = 0.70), and emotional regulation (*M* = 2.82; *SD* = 0.67). 

#### 3.1.3. Demographic Differences in Dependent Variables

There were no demographic differences in the reaction or learning variables. Although there were initial differences in attendance, engagement, and programme enjoyment, these became non-significant after the Benjamini–Hochberg correction. Means, standard deviations, and statistical information can be found in Table 1 and Appendix A. 

### 3.2. Main Analyses 

#### 3.2.1. Relationships between Study Variables 

A correlation matrix can be found in Table 2. The largest relationship was between programme and facilitator evaluation. Other key findings from Table 2 include positive relationships between engagement and reaction variables (i.e., programme evaluation, facilitator evaluation, and programme enjoyment), indicating that higher levels of engagement were associated with more favourable reactions to MST4Life™. These same relationships were not evident for attendance. Additionally, mental skills were positively associated with programme reactions (i.e., programme evaluation, facilitator evaluation, and programme enjoyment) and transfer intention. In other words, greater perceptions of mental skills developed in MST4Life™ were associated with more favourable reactions to the programme and greater intentions to transfer these skills after the programme. 

#### 3.2.2. Reaction Variables Predicting Learning Outcomes 

For all regressions, demographics were entered in Step 1 to account for any confounding variables. Reaction variables were entered in Step 2. All demographic variables at Step 1 were non-significant, and were therefore removed and the regression was rerun, resulting in six linear regressions. There was no evidence of multicollinearity, as all tolerance values were above 0.10 and all VIF values were below 10. The group-work regression was not significant and, therefore, is not presented here. 

All results presented were significant after Benjamini–Hochberg correction, and can be found in Table 3. Programme enjoyment positively predicted transfer intention and all mental skills, and was the strongest predictor for problem solving. Facilitator evaluation also positively predicted transfer intention. Attendance, engagement, and programme evaluation did not significantly predict learning.

#### 3.2.3. Mediation

As programme enjoyment was a key predictor of learning outcomes, further mediation analyses were conducted to explore the potential mechanisms underpinning this relationship. Transfer intention was a significant mediator between programme enjoyment and mental skills: goal setting (B = 0.18, 90% CI = 0.08 to 0.30, completely standardised indirect effect = 0.21), effort (B = 0.12, 90% CI = 0.01 to 0.23, completely standardised indirect effect = 0.15), problem solving (B = 0.16, 90% CI = 0.07 to 0.26, completely standardised indirect effect = 0.20), time management (B = 0.19, 90% CI = 0.08 to 0.32, completely standardised indirect effect = 0.23), emotional regulation (B = 0.23, 90% CI = 0.13 to 0.36, completely standardised indirect effect = 0.27), and group work (B = 0.21, 90% CI = 0.13 to 0.32, completely standardised indirect effect = 0.31). In other words, young people with higher programme enjoyment scores perceived that they had better experiences of mental skills development in MST4Life™, in part through their intentions to transfer the mental skills that they had developed once the programme had finished.

## 4. Discussion

The aim of this process evaluation, as part of a larger evaluation of MST4Life™, was to investigate (1) reactions (i.e., programme evaluation, enjoyment, facilitator evaluation, attendance, and engagement) to and learning from the programme (i.e., mental skills used and transfer intention), (2) the relationships between reaction and learning variables, and (3) transfer intention as a mediator underpinning this relationship. Although research has used the Kirkpatrick model to underpin evaluations outside of business [25,26], to the best of our knowledge, this is the first study to use the Kirkpatrick model in a youth homeless context. This study also extends the scant literature on process evaluations in young people experiencing homelessness and, uniquely, measures programme engagement in addition to attendance. Altogether, this study supports the use of the Kirkpatrick model to evaluate programmes in this context, providing a framework to allow consistency across interventions and, thus, provide clearer recommendations on programme design, delivery, and evaluation for researchers, policymakers, and programme commissioners. Practical suggestions are embedded throughout the discussion to provide recommendations for these groups to improve practice and policy with young people experiencing homelessness, and with disadvantaged youth more broadly.

In line with the first hypothesis, young people had favourable reactions to and learning from MST4Life™, likely due to the strengths-based nature of taking part in a PYD programme [3]. Programme enjoyment was the only significant predictor of mental skills experienced in MST4Life™. This finding partially aligns with research using the Kirkpatrick model, where Cooley et al. [25] evaluated reactions to and learning from an OAE course for university students, and found course enjoyment to be the most significant predictor of course evaluation. Facilitating enjoyment aligns with the SDT underpinning of MST4Life™, and is a key input in the logic model (Figure 1) [8], where consultations with young people and staff emphasised that the programme should be challenging and meaningful, but also fun. Together with qualitative process evaluations of MST4Life™ [12,14], the present research highlights that young peoples’ experiences in programmes (i.e., their enjoyment) should be captured in addition to their learning outcomes so as to understand more about programme effectiveness. This may be particularly pertinent for young people experiencing homelessness, and for disadvantaged youth more broadly, where preferences indicate that avoiding a school-like environment facilitates engagement and learning [14].

Young people also had positive reactions to programme facilitators (*M* = 4.57 out of 5), which correlated with other reaction variables, mental skills experienced (except emotion regulation), and transfer intention. In their pilot life skills program with young people experiencing homelessness, Sisselman-Borgia [19] found that participants noted relationships with programme facilitators and mentors as extremely influential and impactful on their development throughout the programme. Similarly, Sofija et al. [49] highlighted the importance of facilitators in participants’ wellbeing in a group fitness intervention with adults who had experienced homelessness. Rapport development and satisfaction of basic psychological needs were core components of MST4Life™’s delivery style [11,15], and aligned with PYD principles in promoting opportunities to build positive relationships with adults and peers [3]. This quantitative process evaluation complements qualitative evaluations of MST4Life™ that also support the importance of rapport development, a psychologically informed delivery style, and nurturing a sense of belonging [11,12,15]. However, this study extends these findings by exploring how these programme experiences relate to learning outcomes (e.g., mental skills experienced). It is therefore recommended that PYD programmes with disadvantaged youth evaluate not only learning outcomes, but also facilitator delivery style, and how this links to the outcomes experienced.

A novel contribution of this research is measuring programme engagement and exploring its relationship with learning outcomes. Participant engagement has previously been linked to positive outcomes [28,29,30], but has rarely been considered in this population. In MST4Life™, facilitator ratings of young people’s engagement were high (*M* = 8.24 out of 10) and, in contrast to attendance, were associated with more favourable reactions. Promoting engagement was actively considered throughout MST4Life™, where facilitators worked closely with young people and staff to determine what strategies would work best at each accommodation site (e.g., afternoon vs. evening sessions; text reminders vs. knocking on doors; providing autonomy with breaks during sessions). Interventions with adults experiencing homelessness have provided extrinsic rewards for taking part (e.g., stipends based on attendance) [19], whereas engagement in MST4Life™ provided an important indicator of intrinsic motivation [11,12].

In contrast to our hypothesis, neither engagement nor attendance was associated with learning outcomes. However, as part of an overall mixed-methods evaluation, such nuances were captured by a realist evaluation, demonstrating that as young people experienced improvements in wellbeing throughout MST4Life™, this caused a greater shift towards intrinsically motivated reasons for engagement, which led to psychosocial skill development (i.e., learning) later in the programme [12]. It is possible that the quantitative engagement measure did not predict learning outcomes, as this was limited to facilitators’ subjective perceptions. Future research could combine this measure with young people’s own perceptions of their engagement to triangulate such data. Altogether, it is recommended that engagement, as well as attendance, should be included in the delivery and evaluation of programmes with young people experiencing homelessness so as to better understand what constitutes meaningful programme experiences.

In partial agreement with Hypothesis 3, transfer intention mediated the relationship between enjoyment and mental skills developed in MST4Life™. In their literature review of knowledge transfer, Day and Goldstone [33] concluded that transfer is more likely when it is made explicit how transfer can occur in new settings. In MST4Life™, explicit and relevant opportunities for transfer were intentionally created. For example, mental skills developed during earlier sessions were reflected on in terms of how skills could be implemented in future sessions (e.g., time management for planning and running a cake sale on university campus). In the context of the wider evaluation of MST4Life™, young people’s intentions to transfer led to them using these new mental skills on the OAE residential course, which resulted in behavioural changes away from MST4Life™ (e.g., better time management in daily life), as observed by support workers [12]. Altogether, these findings also provide some support for the programme’s logic model, where learning transfer is an expected outcome [8] (Figure 1). It is recommended that explicit opportunities for skill transfer should be embedded within programmes for young people experiencing homelessness, but these opportunities should also continue once the programme has ended, so as to encourage transfer to other contexts (e.g., EET) [12,23].

An original contribution and strength of this study was using the New World Kirkpatrick model [20] to underpin research with young people experiencing homelessness, which provided a structure to explore the links between experiences in the programme and learning outcomes. Employing the Kirkpatrick model in this context emphasised the importance of enjoyment in predicting learning; however, reaction variables were focused on participants’ experiences. It is possible that other reaction variables could further explain the link between reaction and learning. For example, in their model of optimal learning and transfer, Cooley et al. [23] proposed that the characteristics of the environment also influence learning. In the context of programmes within this population, this could include perceptions and accessibility of the location, as well as the extent to which participants are comfortable with the group environment. Furthermore, only levels 1 and 2 (i.e., reaction and learning) of the Kirkpatrick model were included in this study. Moreau [24] notes that most evaluators stop at level 2, and previous research has also focused on these initial levels [25]. To evaluate other levels of the Kirkpatrick model, follow-up data are required; however, given the transient nature of living conditions for this population, these can often be difficult to collect [5]. Although not reported here, higher levels of the Kirkpatrick model have been evaluated through qualitative methods and a cost–benefit analysis for MST4Life™ [12,16]. From our experience working with this population, we recommend that when using the Kirkpatrick model in this context, evaluation indicators should be identified together with collaborators prior to the research in order to agree on the most appropriate methodological approach to obtain the richest data possible.

Although the wider MST4Life™ programme has engaged young people who are underrepresented in research [5], it is not without limitations. The programme facilitators were also those who collected the data. Although internal evaluators may cause social desirability, steps were taken to minimise such biases, such as welcoming both positive and negative views (e.g., the original programme name was changed) [8]. Additional data collection methods were also used, where young people could share their views without the facilitators present (e.g., diary room), and the views of other collaborators were considered (e.g., support workers) [12]. In community-based research, it has been encouraged to use internal evaluators to build relationships between collaborators [5]. In MST4Life™, rapport development between young people and facilitators was crucial to the programme’s success [11]. This rapport also ensured that the young people understood what informed consent was and how their data would be used. Thus, the housing service and researchers agreed that the use of internal evaluators outweighed the strengths of using external evaluators, who young people may have found difficult to trust and openly share their views with, which could have led to disengagement with data collection [21]. Future research evaluating programmes with this population should consider the advantages and disadvantages of using internal or external evaluators, and ensure open conversations between researchers and relevant collaborators (e.g., young people, support workers) to determine the most suitable approach.

Another limitation of the present study is that these results are based on the delivery and evaluation of MST4Life™ within the context of a West-Midlands-based (UK) housing service. The heterogeneity of demographics and support needs of young people experiencing homelessness has been well acknowledged [2,9]. Therefore, future research is required to test the programme’s logic model and its scalability up and out to other housing services, but also to other contexts including young people with multiple disadvantages more broadly [8].

## 5. Conclusions

In conclusion, this process evaluation provides evidence that young people taking part in MST4Life™ had favourable reactions to the programme, which were associated with higher levels of engagement. As programmes typically only measure attendance, engagement should also be included in the delivery and evaluation of programmes with disadvantaged groups to better understand what constitutes meaningful programme experiences. This study also found that programme enjoyment was a key driver of predicting the mental skills experienced, which was mediated by transfer intention. These findings have implications for researchers, housing services, and programme commissioners, as explicit opportunities for skill transfer should be embedded during and after programmes to encourage transfer to other contexts (e.g., EET) for young people experiencing homelessness, and for disadvantaged youth more broadly.

## Figures and Tables

**Figure 1 ijerph-19-11320-f001:**
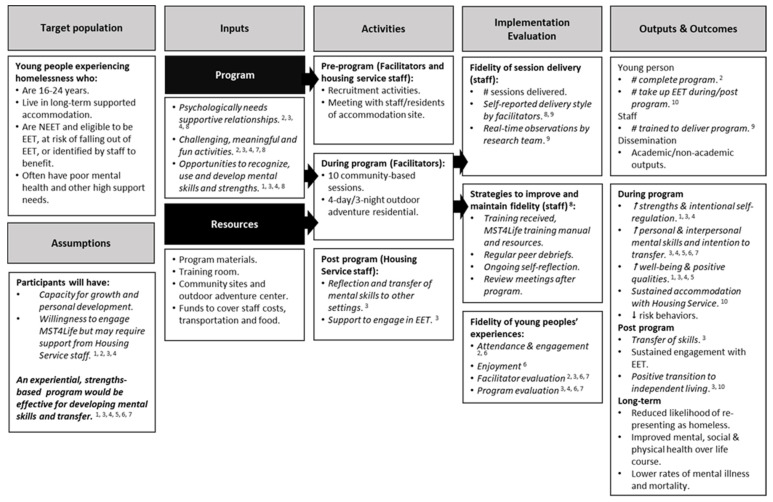
Updated MST4Life™ logic model adapted from Cumming, Whiting et al. (2022) with permission. *Note:* Evidence to support components of the logic model to date is italicised. ^1^ [10], ^2^ [11], ^3^ [12], ^4^ [13], ^5^ [9], ^6^ current manuscript, ^7^ [14], ^8^ [8], ^9^ [15], ^10^ [16].

**Table 1 ijerph-19-11320-t001:** Demographic breakdown of study variables (and standard deviations).

Demographics	Attendance	Engagement	Program Evaluation	Facilitator Evaluation	Program Enjoyment	Transfer Intention
Gender						
Male (*n* = 117)	5.46 (2.68)	8.16 (1.09)	4.30 (0.79)	4.43 (0.73)	4.24 (0.90)	3.85 (0.95)
Female (*n* = 159)	5.33 (2.69)	8.28 (1.15)	4.53 (0.70)	4.67 (0.64)	4.43 (0.72)	4.08 (0.83)
Transgender (*n* = 3)	7.33 (3.79)	9.46 (0.22)	5.00 (0)	5.00 (0)	4.50 (0.71)	4.00 (1.41)
Non-binary (*n* = 1)	6.00 (0)	8.50 (0)	5.00 (0)	5.00 (0)	4.75 (0)	3.50 (0)
Ethnicity						
White (*n* = 154)	5.55 (2.74)	8.34 (1.02)	4.45 (0.71)	4.61 (0.66)	4.32 (0.81)	3.86 (0.92)
Asian/Asian British (*n* = 10)	4.00 (2.54)	7.37 (1.51)	3.75 (1.77)	4.50 (0.71)	4.00 (1.41)	4.00 (1.41)
Black/African/Caribbean/Black British (*n* = 57)	5.17 (2.72)	8.22 (1.23)	4.44 (0.79)	4.60 (0.71)	4.39 (0.76)	4.20 (0.77)
Arab (*n* = 1)	3.00 (0)	8.67 (0)	-	-	-	-
Mixed/multiple ethnic groups (*n* = 46)	5.53 (2.52)	8.11 (1.23)	4.54 (0.85)	4.54 (0.80)	4.56 (0.78)	4.38 (0.84)
Other (*n* = 3)	5.67 (4.04)	8.82 (1.40)	5.00 (0)	5.00 (0)	5.00 (0)	4.00 (0)
Social inclusion						
EET (*n* = 99)	5.54 (2.66)	8.47 (1.13)	4.66 (0.54)	4.79 (0.50)	4.55 (0.60)	4.24 (0.75)
NEET looking for work (*n* = 78)	5.22 (2.65)	8.28 (1.17)	4.15 (0.90)	4.48 (0.79)	4.01 (0.92)	3.64 (0.99)
NEET not looking for work (*n* = 30)	5.32 (3.08)	8.19 (0.78)	4.42 (0.63)	4.54 (0.66)	4.69 (0.45)	4.11 (0.94)
Unable to work/other (*n* = 52)	5.41 (2.84)	7.89 (1.19)	4.45 (0.87)	4.45 (0.78)	4.24 (0.97)	3.83 (0.87)
Learning difficulty						
Yes (*n* = 31)	6.52 (2.62)	8.35 (0.85)	4.50 (0.73)	4.57 (0.68)	4.28 (0.91)	3.70 (1.36)
No (*n* = 154)	5.25 (2.88)	8.46 (1.10)	4.46 (0.75)	4.68 (0.60)	4.48 (0.69)	4.15 (0.77)
Prefer not to say (*n* = 16)	4.75 (2.27)	8.13 (1.03)	4.25 (1.19)	4.38 (1.25)	3.75 (1.34)	3.75 (0.46)

**Table 2 ijerph-19-11320-t002:** Correlation matrix of study variables and Cronbach alphas.

Variables	1.	2.	3.	4.	5.	6.	7.	8.	9.	10.	11.
1. Attendance	-										
2. Engagement	0.15 *	-									
3. Course evaluation	0.15	0.24 *	0.80								
4. Facilitator evaluation	0.14	0.25 **	0.81 ***	0.76							
5. Enjoyment	0.09	0.25 **	0.75 ***	0.59 ***	0.95						
6. Transfer intent	0.06	0.17	0.56 ***	0.55 ***	0.57 ***	0.92					
7. Goal setting	−0.04	0.04	0.38 ***	0.26 **	0.42 ***	0.47 ***	0.83				
8. Effort	0.03	0.11	0.34 ***	0.26 **	0.46 ***	0.42 ***	0.68 ***	0.82			
9. Problem solving	0.07	0.15	0.34 ***	0.27 **	0.50 **	0.49 ***	0.63 ***	0.64 ***	0.82		
10. Time management	0.07	0.06	0.35 ***	0.24 *	0.41 ***	0.48 ***	0.64 ***	0.69 ***	0.68 ***	0.82	
11. Emotional regulation	−0.05	−0.04	0.23 *	0.15	0.33 ***	0.51 ***	0.52 ***	0.53 ***	0.53 ***	0.69 ***	0.76
12. Group work	−0.05	0.17	0.36 ***	0.36 ***	0.35 ***	0.56 ***	0.56 ***	0.53 ***	0.59 ***	0.58 ***	0.58 ***

Note: * *p* < 0.05, ** *p* < 0.01, *** *p* < 0.001. Cronbach alphas are reported on the diagonal.

**Table 3 ijerph-19-11320-t003:** Linear regressions for reaction predicting learning variables.

Reaction Variables	*B*	*SE B*	β	*t*	*p*	*R^2^ (Cohen’s f^2^)*	*Sig*
Transfer Intention							
						0.41 (0.69)	<0.001 ***
Attendance	−0.01	0.03	−0.02	−0.27	0.787		
Engagement	0.00	0.07	0.00	0.00	0.999		
Programme evaluation	0.05	0.19	0.05	0.28	0.780		
Facilitator evaluation	0.40	0.18	0.30	2.21	0.029 *		
Enjoyment	0.40	0.13	0.37	3.11	0.002 **		
Goal-setting							
						0.21 (0.27)	<0.001 ***
Attendance	−0.02	0.03	−0.06	−0.70	0.488		
Engagement	−0.05	0.08	−0.06	−0.61	0.544		
Programme evaluation	0.21	0.17	0.22	1.22	0.224		
Facilitator evaluation	−0.10	0.16	−0.10	−0.65	0.518		
Enjoyment	0.30	0.12	0.35	2.60	0.011 *		
Effort							
						0.23 (0.30)	<0.001 ***
Attendance	0.00	0.03	0.00	−0.01	0.989		
Engagement	0.00	0.07	0.00	−0.03	0.979		
Programme evaluation	−0.02	0.16	−0.03	−0.15	0.879		
Facilitator evaluation	0.00	0.15	0.00	−0.05	0.959		
Enjoyment	0.40	0.11	0.50	3.74	0.000 ***		
Problem solving							
						0.26 (0.35)	<0.001 ***
Attendance	0.01	0.03	0.04	0.41	0.681		
Engagement	0.02	0.07	0.02	0.22	0.829		
Programme evaluation	−0.09	0.16	−0.10	−0.59	0.559		
Facilitator evaluation	0.03	0.15	0.03	0.19	0.848		
Enjoyment	0.45	0.11	0.56	4.24	0.000 ***		
Time management							
						0.19 (0.23)	0.001 **
Attendance	0.02	0.04	0.05	0.54	0.593		
Engagement	−0.04	0.08	−0.06	−0.57	0.572		
Programme evaluation	0.16	0.17	0.18	0.96	0.342		
Facilitator evaluation	−0.08	0.16	−0.08	−0.51	0.615		
Enjoyment	0.29	0.12	0.34	2.50	0.014 *		
Emotional regulation							
						0.14 (0.16)	0.010 *
Attendance	−0.02	0.04	−0.05	−0.55	0.585		
Engagement	−0.06	0.08	−0.08	−0.82	0.415		
Programme evaluation	0.03	0.17	0.03	0.15	0.879		
Facilitator evaluation	−0.08	0.16	−0.08	−0.52	0.603		
Enjoyment	0.34	0.12	0.40	2.73	0.008 **		

Note: * *p* < 0.05, ** *p* < 0.01, *** *p* < 0.001. Cronbach alphas are reported on the diagonal.

## Data Availability

Data not available. Due to ethical concerns, participants were assured that the raw data would remain confidential and would not be shared.

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
