# Peer review of "A Kirkpatrick Model Process Evaluation of Reactions and Learning from My Strengths Training for Life™"

_ijerph, 2022, doi:10.3390/ijerph191811320_

Round 1

Reviewer 1 Report

First of all, thank you for inviting me to review the paper entitled “A Kirkpatrick model process evaluation of reactions and learning from My Strengths Training for Life”.

Paper is somewhat interesting, wherein the Kirkpatrick model was used to evaluate the effectiveness/outcomes of a community based sport psychology program for young people (My Strengths Training for Life). Paper is well written and methodology clear and sound.

Some suggestions and recommendations are provided:

1.      Introduction is quite long, would suggest to provide – subheadings or subsection, eg line 98 – the new world Kirkpatrick model

2.      Sampling? Inclusion / exclusion criteria?

3.      Why these control variables? (hierarchical linear regressions), effect sizes of the results

4.      Line 293 … Process which model? (please indicate for clarity, thanks)

5.      Why the need for Benjamini-Hochberg correction (line 348), for better clarify.

6.      Is there a need for confirmatory factor analysis?

7.      For the section 3.2.3 – mediation, would recommend to provide more details, perhaps a table and a conceptual diagram or structural diagram, how about Bootstrap? Total effects? So its partial mediation? Or full? – please clarify, thanks

8.      Do expand your implications to include more practical suggestions

In sum, the study is well conducted and detailed with its methodology, just some minor clarifications in the results section, although the focus is on the findings, however, clarity of the statistical (mediation section) measures should be provided.

Reviewer 2 Report

A very good paper - For consideration, I recommend the inclusion of limitations more explicitly in the paper that would enable the reader audience to understand any limitations and any caveats since doing so will enable reader(s) have the opportunity to place the paper into a context in relation to what is stated / proposed by the author. In my view, this minor revision would take the paper to a level that draws readership and a following for your research.

Overall, an interesting and engaging paper, which investigated Kirkpatrick model in the context of a community-based, sport psychology program (My Strengths Training for Life™) for young people experiencing homelessness via (1) young peoples’ reactions (program and facilitator evaluation, enjoyment, attendance, and engagement) to and learning (mental skills and transfer intention), (2) the relationship between reaction and learning variables, and (3) the mediators underpinning this relationship with a population of 301 young people living in a West Midlands housing service within the UK.

Given how this research is focused on youth and homelessness, the paper will garner interest from academia, and NGO and government policy makers. From an academic perspective the mix of current (less than 5 year peer reviewed research) presented and discussed in the paper is per se robust in terms of currency; and there is a very good discussion of discussion of divergent and conflicting perspectives from the literature evident, which will scholars expect in journal articles. As well, and on a positive note, the author(s) has provided some germane discussion with the literature. From a policy perspective, both NGO and government policy makers will view the paper, and its findings relevant in the context of youth homelessness since it identifies explicit opportunities for skill transfer should be embedded during and after programs to encourage transfer to other contexts for young people experiencing homelessness and disadvantaged youth more broadly, which can drive shifts in policy which are need to support change.

From a methodology perspective, the paper is well designed and appropriate – the exception is the lack of limitations noted explicitly in the paper, and this may very well reflect my own personal worldview. However, in defense of this posture, the identification of limitations more explicitly in the paper would enable the reader audience to understand the limitations and any caveats. For this reason, it is imperative that reader(s) have the opportunity to place the paper into a context in relation to what is stated / proposed by the author.

The results are presented in a clear and concise manner; there is evidence of analysis evident; the inclusion of Tables is an effective visual. As well, the author(s) have developed an acceptable linkage between the results and conclusions noted. The points noted in paper are tied together into a final coherent picture. It is evident that the author(s) have an excellent understanding of the subject area.

This is a solid paper in many respects, since it provides several opportunities for continued research in the subject area with the possibility of different streams within the research area, while providing further avenues of research potential. With respect to the practical application of the research, it presents an opportunity to enhance the depth, breadth and understanding of the explicit opportunities for skill transfer that should be embedded during and after programs to encourage transfer to other contexts for young people experiencing homelessness and disadvantaged youth more broadly, which can drive shifts in policy which are need to support change and how NGO and government policy makers can tailor strategies to enhance their effectiveness.

The paper is well written with excellent readability and scholarly flow. There is no need for any further English language review.
